# Ethephon Treatment Enhanced Postharvest Litchi Fruit Resistance to *Peronophythora litchii* by Strengthening Antioxidant Capacity and Defense Systems

**DOI:** 10.3390/foods14203493

**Published:** 2025-10-14

**Authors:** Difa Zhu, Tao Luo, Xiaomeng Guo, Jingyi Li, Qiao Li, Yongqi Chen, Wenbo Ou, Dongmei Han, Zhenxian Wu

**Affiliations:** 1Guangdong Provincial Key Laboratory of Postharvest Science of Fruits and Vegetables, Engineering Research Center of Southern Horticultural Products Preservation, Ministry of Education, College of Horticulture, South China Agricultural University, Guangzhou 510642, China; dfzhu@stu.scau.edu.cn (D.Z.); luotao0502@scau.edu.cn (T.L.); guoxm_scau@163.com (X.G.); jyli0520@163.com (J.L.); liqiao@stu.scau.edu.cn (Q.L.); winky0428@stu.scau.edu.cn (Y.C.); xboo118@163.com (W.O.); 2Key Laboratory of Biology and Genetic Improvement of Horticultural Crops (South China), Ministry of Agriculture and Rural Affairs, College of Horticulture, South China Agricultural University, Guangzhou 510642, China; 3Guangxi Key Laboratory of Health Care Food Science and Technology, School of Food and Biological Engineering, Hezhou University, Hezhou 542899, China; 4Key Laboratory of South Subtropical Fruit Biology and Genetic Resource Utilization, Ministry of Agriculture and Rural Affairs, Guangdong Provincial Key Laboratory of Science and Technology Research on Fruit Tree, Institute of Fruit Tree Research, Guangdong Academy of Agricultural Sciences, Guangzhou 510640, China; handongmei@gdaas.cn

**Keywords:** litchi fruit, *P. litchii*, ethephon, antioxidant, defense, pericarp browning, disease resistance

## Abstract

Litchi downy blight, caused by *Peronophythora litchii*, is a major postharvest disease that leads to severe pericarp browning and fruit decay, significantly reducing market quality. Strengthening the fruit’s innate defense systems represents a promising strategy for minimizing these losses. This study investigated the efficacy and underlying mechanisms of ethephon treatment in controlling postharvest litchi downy blight. The results showed that treatment with 400 mg·L^−1^ ethephon solution via a 2-min immersion significantly suppressed *P. litchii* infection, reduced the disease index and pericarp browning index, and enhanced the rate of ethylene production. Ethephon application notably increased 2,2-diphenyl-1-picrylhydrazyl (DPPH) scavenging activity, and the activities of key antioxidant and defense-related enzymes, including superoxide dismutase (SOD), catalase (CAT), ascorbate peroxidase (APX), chitinase (CHI), β-1,3-glucanase (GLU), and phenylalanine ammonia lyase (PAL). Concurrently, it up-regulated the expression of corresponding genes *LcCAT*, *LcAPX*, *LcCHI*, *LcGLU*, *LcPAL*. In contrast, ethephon treatment reduced the accumulation of hydrogen peroxide (H_2_O_2_) and malondialdehyde (MDA). In summary, ethephon treatment suppresses postharvest litchi downy blight likely through the enhancement of both antioxidant and pathogen defense capacities. These findings provide valuable insights into the potential application of ethephon for maintaining postharvest quality in litchi fruit.

## 1. Introduction

Litchi (*Litchi chinensis* Sonn.) is highly competitive in the global fruit market due to its unique flavor and high nutritional value [1]. As the world’s largest producer and consumer of litchi, China recorded a production exceeding 3 million tons in both 2018 and 2023 [2,3]. However, the hot and humid conditions typical of the concentrated harvest period promote the proliferation of postharvest diseases, leading to rapid fruit decay and browning that severely restrict the market availability window of fresh litchi fruits [4]. Among these diseases, litchi downy blight, caused by *Peronophythora litchii* (a hemibiotrophic oomycete pathogen), is one of the most destructive, accounting for approximately 20–30% of annual yield losses [5]. Currently, sulfur dioxide (SO_2_) fumigation and the application of chemical fungicides remain the primary control measures against litchi downy blight [6,7]. However, these methods raise concerns regarding environmental pollution and food safety. Consequently, the use of alternative treatments with natural antimicrobial activity or induced resistance has emerged as a research focus in recent years. Examples include biocontrol fungal preparations [8,9,10], plant extracts [11,12,13], and exogenous phytohormones [4,14].

The induction of systemic resistance (ISR) to potentiate the intrinsic defense capacity of fruit has emerged as a sustainable and effective phytosanitary strategy against postharvest diseases [15,16]. In the case of litchi downy blight, a conserved mechanistic basis for this induced resistance has been elucidated. Studies have demonstrated that treatment with diverse elicitors, including *Exiguobacterium acetylicum* Strain SI17 [5], *Bacillus amyloliquefaciens* PP19 [8], kadozan [17], 6-pentyl-2H-pyran-2-one [18] and benzothiadiazole [19], triggers a rapid potentiation of the host’s antioxidant system. This is characterized by the upregulation of key ROS clearance enzyme activities, including superoxide dismutase (SOD), catalase (CAT), peroxidase (POD), and ascorbate peroxidase (APX), alongside an accumulation of non-enzymatic antioxidants, which collectively mitigate phytotoxic oxidative burst by reducing hydrogen peroxide (H_2_O_2_) accumulation and membrane lipid peroxidation. Concurrently, a parallel defensive layer is activated through the induction of defense-related proteins. The synergistic action of chitinase (CHI) and β-1,3-glucanase (GLU) directly targets and degrades essential structural components of the pathogen’s cell wall, while phenylalanine ammonia-lyase (PAL) drives the phenylpropanoid pathway for the accumulation of antimicrobial compounds [11,17]. Thus, the elicited resistance against *P. litchii* is orchestrated through a multifaceted network, integrating antioxidative protection with direct antimicrobial strategies. However, all currently known induction treatments exhibit certain limitations that have prevented their large-scale application in production.

Ethephon, a widely utilized commercial ethylene-releasing agent, plays a significant role in modulating key physiological processes in plants, including growth regulation, fruit ripening, senescence, and disease resistance [20,21,22]. It has been demonstrated to enhance disease resistance across a range of plant species, including apple [23], melon [24], cucumber [25], pepper [26], wheat [27], tomato [28], mandarins [29], and grape [30]. Mechanistically, studies have demonstrated that exogenous ethephon application activates the phenylpropanoid pathway in apple leaves [23] and grape berries [30], promoting the biosynthesis of flavonoids and anthocyanins. In wheat, ethephon mitigates *Stagonospora nodorum*-induced H_2_O_2_ accumulation by enhancing CAT activity [27]. Similarly, transient overexpression of the ethylene-responsive gene *CYP82D47* in cucumber leaves enhances the activities of CAT and SOD, thereby improving resistance to *Fusarium* wilt [31]. In tomato fruit, the activities of CHI and GLU exhibit a strong correlation with ethylene release and contribute to resistance against *Botrytis cinerea* [32,33]. Ethylene treatment also activates the expression of key genes *VvChi5/17/22/37* in both grape leaves and berries, which are associated with resistance to *Botrytis cinerea* [34]. Furthermore, ethylene-mediated regulation of antioxidant capacity and defense-related enzyme activities has been reported during postharvest storage in Chinese water chestnuts [35], cassava [36], cactus pear [37], Anxi persimmons [38], and banana [39].

However, it remains unclear whether ethephon can be effectively applied to control postharvest litchi downy blight. Therefore, this study investigated the inhibitory effect of ethephon against *P. litchii* and its mechanism of inducing the resistance in litchi fruit.

## 2. Materials and Methods

### 2.1. Fruit Material and P. litchii

Mature litchi fruits (cultivar ‘Huaizhi’) were collected in the morning from the Litchi Exposition Park in Guangzhou, Guangdong, China. After transport to the laboratory within three hours of harvest, healthy fruits with uniform size were selected and washed for the experiment.

The *P. litchii* isolate was provided by the College of Plant Protection, South China Agricultural University, and cultured on carrot agar media (200 g·L^−1^ carrot juice, 20 g·L^−1^ agar) at 25 °C for five days [40]. Following incubation, sporangia were eluted with sterile water and filtered to remove detached mycelia. A suspension adjusted to a concentration of 1 × 10^4^ sporangia per mL was then prepared for inoculation.

### 2.2. Treatments

The selected litchi fruits were disinfected using a 450 mg·L^−1^ prochloraz solution for 2 min, immediately rinsed with water, and air-dried [19]. Approximately 600 healthy litchi fruits were divided into two groups and treated as follows: (1) Control, soaked in sterile water for 2 min; (2) Ethephon, soaked in 400 mg·L^−1^ ethephon solution for 2 min. After air-drying, all fruits were inoculated with 20 μL of a 1 × 10^4^ mL^−1^ *P. litchii* sporangia suspension. Subsequently, the fruits were packaged in 0.02 mm polyvinyl chloride bags and stored at room temperature (25 ± 1 °C) for 96 h. Three biological replicates were established for every group of 20 fruits. Observations and sampling were conducted at 0, 24, 48, 72, and 96 h after inoculation. Non-necrotic pericarp tissues were snap-frozen in liquid nitrogen and stored at −80 °C for subsequent analysis.

### 2.3. Assessment of Disease Index and Pericarp Browning Index

The disease index was evaluated based on methods described by Guo et al. [41] and Jiang et al. [17]. The lesion diameter on the pericarp surface was categorized into 0–6 grades to assess the severity of the fruit disease, with the scoring criteria as follows: Grade 0, no disease; Grade 1, 0 cm < lesion diameter ≤ 0.5 cm; Grade 2, 0.5 cm < lesion diameter ≤ 1 cm; Grade 3, 1 cm < lesion diameter ≤ 2 cm; Grade 4, 2 cm < lesion diameter ≤ 3 cm; Grade 5, 3 cm < lesion diameter ≤ 4 cm; and Grade 6, lesion diameter > 4 cm. The disease index for each replicate was calculated as follows:(1)$$Disease\;index=\frac{\sum (Disease\;grade\times Number\;of\;fruits\;in\;each\;grade)}{Total\;number\;of\;fruits}$$

The pericarp browning index was evaluated according to the method of Guo et al. [42].

### 2.4. Measurement of Ethylene Production Rate

Eight fruits were placed in 600 mL sealed boxes with rubber stoppers, and three biological replicates were set up for each group. The gas to be measured was collected using 1 mL syringes after sealing for 3 h at 25 °C. Ethylene production rate was quantified using a Shimadzu GC-2014C gas chromatograph (Shimadzu Corporation, Kyoto, Japan) equipped with a flame ionization detector and an activated alumina column (200 cm × 0.3 cm) and expressed as μL·kg^−1^·h^−1^.

### 2.5. Determination of Reactive Oxygen Species (ROS) Levels, 2,2-Diphenyl-1-Picrylhydrazyl (DPPH) Scavenging Activity and Malondialdehyde (MDA) Contents

The superoxide anion (O_2_^•−^) production rate was determined using the hydroxylamine oxidation method [43]. A 2 g sample of pericarp tissue was homogenized with 10 mL of ice-cold 50 mM phosphate-buffered saline (PBS, pH 7.5) containing 1 mM ethylenediaminetetraacetic acid (EDTA), 0.3% (*v*/*v*) Triton X-100, and 2% (*w*/*v*) Polyvinylpolypyrrolidone (PVPP), with the supernatant collected after centrifugation. A 0.6 mL aliquot of extraction solution was mixed with 0.6 mL of 50 mM PBS (pH 7.5) and 0.6 mL of 1 mM hydroxylamine hydrochloride. The mixture was incubated at 25 °C for 1 h. Then, 0.6 mL of 17 mM sulfanilic acid and 0.6 mL of 7 mM α-naphthylamine were added, followed by another 20 min of incubation at 25 °C. The absorbance of the final solution was measured at 530 nm. The control measurement was performed following the same procedure but omitting the 1 h incubation step. The O_2_^•−^ production rate was expressed as pmol·min^−1^·g^−1^.

H_2_O_2_ content was measured according to the titanium sulfate precipitation method [43]. A 1 g sample of litchi pericarp was homogenized in 5 mL of pre-chilled acetone, and the homogenate was centrifuged to obtain the supernatant. An aliquot of 0.8 mL of the supernatant was reacted with 80 μL of 10% TiCl_4_-HCl solution and 80 μL of concentrated NH_4_OH for 5 min. The homogenization was then centrifuged, the supernatant was removed, and the pellet was subjected to multiple washes with acetone (pre-cooled to −20 °C) until complete decolorization was achieved. The purified pellet was subsequently dissolved in 2.4 mL of 2 M H_2_SO_4_, and the absorbance was read at 412 nm, and H_2_O_2_ content was expressed as mmol·g^−1^.

The DPPH scavenging activity of litchi pericarp was evaluated following the protocols of Guo et al. [44]. Briefly, 2 g of pericarp tissue was homogenized with 10 mL of absolute ethanol, and the supernatant was collected after centrifugation. Then, 2 mL of the extraction solution was mixed with 2 mL of 0.1 mM DPPH solution. The mixture was allowed to react in the dark for 30 min, and the absorbance was measured at 517 nm. The control was prepared by replacing the extract with an equal volume of absolute ethanol.

The MDA content was determined by the thiobarbituric acid (TBA) method [45]. Briefly, 2 g of pericarp tissue was homogenized with 10 mL of 10% (*w*/*v*) trichloroacetic acid (TCA). The homogenate was centrifuged to collect the supernatant. Then, 2 mL of the extraction solution (the extraction buffer as control) was mixed with 2 mL of 0.67% (*w*/*v*) TBA solution. The mixture was heated in a boiling water bath for 20 min, cooled, and then its absorbance was measured at 450 nm, 532 nm, and 600 nm. The MDA content was expressed as nmol·g^−1^.

### 2.6. Determination of Antioxidant Enzymes Activities

The activities of SOD, CAT, POD, and APX were determined according to methods described by Guo et al. [44] and Yun et al. [46] with slight modifications.

For the analysis of POD activity, the crude enzyme solution was obtained by homogenizing 2 g of litchi pericarp with 10 mL of ice-cold 50 mM PBS (pH 7.0) containing 5% (*w*/*v*) PVPP and 0.1% (*w*/*v*) EDTA, followed by centrifugation to collect the supernatant. POD activity was measured in a reaction system containing 0.4 mL of enzyme extraction solution, 2 mL of 50 mM PBS (pH 7.0), 0.6 mL of 25 mM guaiacol, and 0.2 mL of 0.5 M H_2_O_2_. The reaction was initiated by rapid mixing, and the increase in absorbance at 470 nm was recorded for 3 min.

For the determination of SOD, CAT, and APX activities, a common crude enzyme extraction solution was prepared by homogenizing 2 g of litchi pericarp with 10 mL of ice-cold 100 mM PBS (pH 7.5) containing 5% (*w*/*v*) PVPP and 5 mM dithiothreitol (DTT), with the supernatant collected after centrifugation. SOD activity was assayed in a reaction mixture containing 0.1 mL of enzyme extraction solution (pure water as control), 1.7 mL of 0.1 M PBS (pH 7.5), 0.3 mL of 130 mM L-methionine, 0.3 mL of 750 μM nitrotetrazolium blue chloride (NBT), 0.3 mL of 0.1 mM EDTA-Na_2_, and 0.3 mL of 20 μM riboflavin. After mixing, the reaction was conducted under 4000 lx illumination for 15 min, terminated by darkness, and absorbance was measured at 560 nm. CAT activity was determined by mixing 0.15 mL of enzyme extraction solution (pure water as control) and 3 mL of 20 mM H_2_O_2_. And the decrease in absorbance at 240 nm was recorded for 3 min. APX activity was assessed in a reaction mixture containing 0.25 mL of enzyme extraction solution, 2.5 mL of 50 mM PBS (pH 7.5) with 0.1 mM EDTA and 0.5 mM ascorbic acid, and 0.25 mL of 2 mM H_2_O_2_. After mixing, the decrease in absorbance at 290 nm was monitored for 3 min.

For the definition of enzyme activity, one unit (U) of CAT, POD, or APX activity was defined as a change in absorbance of 0.1 per minute in the enzymatic reaction system. One unit of SOD activity was defined as the amount of enzyme required to cause 50% inhibition of NBT photoreduction per minute.

### 2.7. Determination of Defense Related Enzymes Activities

The activities of CHI and GLU were measured using the CHI Assay Kit and GLU Assay Kit (Ruixin Biotechnology Co., Ltd., Quanzhou, China), respectively, according to the manufacturer’s instructions. One unit of CHI activity was defined as the production of 0.1 mg N-acetyl-D-(+)-glucosamine equivalents per hour. Similarly, one unit of GLU activity was defined as the production of 1 μmol of glucose equivalents per minute. PAL activity was determined following Wu et al. [47]; one unit was defined as a 0.01 change in absorbance at 290 nm per minute.

### 2.8. RNA Extraction and Gene Expression Analysis

Total RNA was extracted from litchi pericarp using an Ultrafast Plant RNA Extraction Kit (Huayueyang Biotechnology Co., Ltd., Beijing, China) following the manufacturer’s protocol. cDNA was synthesized using the Hifair III 1st Strand cDNA Synthesis SuperMix for qPCR (gDNA digester plus) Kit (Yeasen Biotechnology Co., Ltd., Shanghai, China). Quantitative real-time PCR (qRT-PCR) was performed using the SYBR^®^ Green Premix Pro Taq HS qPCR Kit (Accurate Biology Co., Ltd., Changsha, China) and the Bio-Rad/CFX Touch384 fluorescence quantitative PCR instrument (Bio-Rad Laboratories, Inc., Hercules, CA, USA). Gene sequences used in this study were obtained from the Sapindaceae Genome website [48]. Gene expression levels were normalized to *LcActin* (*HQ615689*) as an internal standard, and relative expression levels of target genes were calculated using the 2^−ΔΔCt^ method [42,49]. Each reaction was performed in triplicate. Primer sequences for the genes are provided in Appendix A.

### 2.9. Statistical Analyses

Data analysis encompassing mean values, standard error, and significance was conducted utilizing an independent samples *t*-test in IBM SPSS Statistics 26. Graphs were generated with Origin 2025, depicting mean values with standard errors (mean ± standard error, n = 3). Additionally, Pearson’s correlation analysis and hierarchical clustering were performed using the Dynamic Correlation Heatmap tool on the Omicsmart online analytics platform to assess the correlations between measured indices.

## 3. Results

### 3.1. Effect of Ethephon Treatment on Postharvest Litchi Downy Blight, Disease Index, Pericarp Browning Index, and the Ethylene Production Rate After Inoculation with P. litchii

Litchi fruits underwent rapid deterioration following inoculation with *P. litchii*, with severe symptoms evident within 96 h. As depicted in Figure 1A, control fruits exhibited extensive black lesions and dense white powdery mildew layers within 72–96 h. In contrast, ethephon treatment significantly suppressed lesion expansion and the development of white powdery mildew layers.

Both the disease index and pericarp browning index increased progressively with the progression of litchi downy blight. Ethephon application significantly reduced the disease index from 24 to 96 h (Figure 1B) and the pericarp browning index at 72 and 96 h compared to the control (Figure 1C).

Notably, infection by *P. litchii* triggered ethylene biosynthesis in litchi fruit. Ethephon treatment further enhanced the ethylene production rate at 24 and 48 h relative to the infected control (Figure 1D).

Collectively, these results indicated that ethephon treatment effectively mitigated the development of litchi downy blight and pericarp browning, while concurrently amplifying the ethylene release in postharvest litchi fruit.

### 3.2. Effect of Ethephon Treatment on the ROS Levels, DPPH Scavenging Activity, and MDA Content in Postharvest Litchi Fruit

Following inoculation with *P. litchii*, both control and ethephon-treated fruits exhibited increasing trends in O_2_^•−^ production rate, H_2_O_2_ content, DPPH scavenging activity, and MDA content (Figure 2).

Ethephon-treated fruits showed a significantly higher O_2_^•−^ production rate compared to the control at 48 and 96 h after inoculation (Figure 2A). Conversely, H_2_O_2_ content in the treated fruits was significantly lower than that in the control from 24 to 96 h after inoculation (Figure 2B).

Meanwhile, the DPPH scavenging activity was significantly enhanced in ethephon-treated fruits throughout the entire observation period (Figure 2C). Additionally, MDA content was significantly reduced in ethephon-treated fruits at 48 and 72 h after inoculation (Figure 2D).

These results suggested that ethephon treatment alleviated oxidative damage in postharvest litchi fruit, as indicated by reduced H_2_O_2_ accumulation and lipid peroxidation.

### 3.3. Effect of Ethephon Treatment on Antioxidant Indices in Postharvest Litchi Fruit

Ethephon treatment exerted complex and dynamic regulatory effects on the antioxidant system in postharvest fruit. Contrary to the overall increasing trend of SOD activity, the expression of the *LcSOD* in control fruits decreased consistently during storage. In comparison to the control, ethephon-treated fruits exhibited a significant yet transient increase in SOD activity at 24 h, followed by a significant lower at 96 h (Figure 3A,E).

CAT activity and *LcCAT* expression in control fruits showed a coordinated increase from 48 to 72 h, followed by a decline. Notably, ethephon treatment significantly enhanced both CAT activity and *LcCAT* expression relative to the control at 24, 48, and 96 h (Figure 3B,F).

Regarding the POD system, both control and treated fruits exhibited a common trend of an initial decrease (0–48 h) followed by an increase. Ethephon treatment led to a time-dependent suppression of the POD system, manifested by significantly lower *LcPOD* expression at 72 h and POD activity at 96 h compared to the control (Figure 3C,G).

APX activity and *LcAPX* expression in control fruits increased in parallel from 48 to 72 h, and then declined synchronously. Ethephon treatment significantly upregulated both APX activity and *LcAPX* expression at 24 h compared to the control (Figure 3D,H).

These results demonstrate that ethephon treatment enhanced the antioxidant capacity of postharvest litchi fruit through modulating the activity and expression of key antioxidant enzymes.

### 3.4. Effect of Ethephon Treatment on Defense-Related Enzymes Activities and Key Gene Expression in Postharvest Litchi Fruit

Upon inoculation with *P. litchii*, the control fruits exhibited increasing activities of CHI and PAL, along with upregulation of *LcCHI*, *LcGLU*, and *LcPAL* expression during storage, whereas GLU activity displayed an initial rise followed by a gradual decline (Figure 4). Ethephon treatment further enhanced these defense-related responses. Specifically, CHI activity and *LcCHI* expression were significantly higher in ethephon-treated fruits at 24 and 72 h, and from 24 to 72 h, respectively, compared to the control (Figure 4A,D). Similarly, both GLU activity and *LcGLU* expression were significantly elevated in the ethephon-treated fruits from 24 to 96 h and from 48 to 96 h, respectively (Figure 4B,E). Additionally, PAL activity and *LcPAL* expression were significantly enhanced at 24 and 48 h but were lower than those in the control at 72 and 96 h after treatment (Figure 4C,F). Together, these results indicated that ethephon treatment enhanced the defense capacity of postharvest litchi fruit by boosting the activity and gene expression of key defense-related enzymes.

### 3.5. Correlation and Hierarchical Cluster Analysis of Physiological and Biochemical Indices in Control and Ethephon-Treated Fruits

Pearson correlation analysis was conducted to evaluate the pairwise correlation between three key indicators (disease index, pericarp browning index, and ethylene production rate) and 21 physiological and biochemical indices in both control and ethephon-treated fruits, respectively (Figure 5). In control fruits, a significant correlation was observed among disease index, pericarp browning index, and ethylene production rate. Moreover, these three indices exhibited significant positive correlations with O_2_^•−^ production rate, H_2_O_2_ content, DPPH scavenging activity, MDA content, SOD activity, APX activity, CHI activity, PAL activity, and the expression of *LcCHI*, *LcGLU*, and *LcPAL*, but were significantly negatively correlated with the expression of *LcSOD* (Figure 5A). In contrast, in ethephon-treated fruits, ethylene production rate was significantly positively correlated not only with *LcCAT* expression and the activities of CAT and GLU, but also exhibited a stronger correlation with CHI activity (Figure 5B). Taken together, these findings indicated that changes in the activities of CAT, GLU, and CHI in response to ethephon treatment exhibited a strong association with ethylene production.

## 4. Discussion

### 4.1. Ethephon Treatment Enhanced Ethylene Signal to Improve Postharvest Litchi Fruit Resistance Against P. litchii Infection

*P. litchii*-induced litchi downy blight is a devastating postharvest disease, characterized by rapid transmission through mycelial proliferation and spore dispersal, which significantly accelerates fruit browning and decay [50,51]. Our investigation revealed a highly significant positive correlation (*p* < 0.001) between disease progression severity and pericarp browning intensity (Figure 1 and Figure 5), indicating a synergistic deterioration pattern. Although various treatments—such as 6-Pentyl-2H-pyran-2-one [18], Ag-NPs [52], benzothiadiazole [19], pterostilbene [7], apple polyphenols [11], kadozan [17], melatonin [4], and multiple bacterial biocontrol agents [53,54,55]—have demonstrated efficacy against this pathogen, their practical application remains limited by scalability and economic feasibility.

Ethephon, as a low-toxicity, economical, and commercially available ethylene-releasing agent, plays a significant role in regulating plant disease resistance [23,24]. However, its application in postharvest litchi fruit preservation has not been previously reported. Our study established that treating postharvest litchi fruit with 400 mg·L^−1^ ethephon effectively suppressed downy blight development and significantly reduced both disease index and pericarp browning index (Figure 1). This concentration was determined through preliminary systematic screening experiments (conducted at 0, 200, 400, and 800 mg·L^−1^) with due consideration for food safety requirements. Notably, this concentration falls within the established effective range (100–1000 mg·L^−1^) reported for ethephon-mediated pathogen control in other postharvest fruits, such as tomato [28], mandarins [29], and grape [30]. Furthermore, existing studies have shown that in postharvest mango and tomato fruits treated with 1000 mg·L^−1^ ethephon (by soaking for 5 min), the residue of ethephon can decrease to below 1 mg·kg^−1^ within 3 days [56,57], which is lower than the maximum residue limit (2 mg·kg^−1^) specified in the National Food Safety Standard of China (GB 2763-2021) for fruits such as mango, tomato, and litchi [58]. Therefore, it can be reasonably concluded that the concentration of ethephon used in this study poses a relatively low risk to food safety.

Mechanistically, ethephon is hydrolyzed within cells to release ethylene, which subsequently plays a role in regulating physiological responses [28,59]. The recent research indicates that ethylene acts as a key positive regulator in conferring resistance against litchi downy blight in litchi leaves: application of 1-aminocyclopropane-1-carboxylic acid (an ethylene biosynthesis precursor) enhances resistance, whereas aminoethoxyvinylglycine (an ethylene biosynthesis inhibitor) compromises defense capacity [60]. In the present study, we provided evidence that ethylene played a functionally consistent role in postharvest litchi fruit. Specifically, a significant positive correlation was observed between ethylene production rate and both disease index and pericarp browning index in control fruits, indicating its critical role in the response to *P. litchii* infection (Figure 5). More importantly, ethephon-treated litchi fruits released substantial ethylene and exhibited higher resistance to litchi downy blight (Figure 1).

Collectively, this study demonstrated that ethephon mitigated postharvest fruit deterioration caused by litchi downy blight by modulating ethylene-mediated defense mechanisms, highlighting its potential as an effective and economically feasible strategy for maintaining postharvest litchi fruit quality.

### 4.2. Ethephon Treatment Inhibited the Development of Litchi Downy Blight by Enhancing Antioxidant Capacity of Postharvest Litchi Fruit

Pathogen-induced ROS burst plays a dual role in plant defense: it can damage fungal hyphae while also activating host defense responses [15]. However, excessive ROS accumulation causes oxidative damage to plant cellular membranes and may paradoxically promote pathogen invasion [61]. Therefore, maintaining appropriate ROS scavenging capacity is crucial for preserving membrane integrity and enhancing resistance to pathogens [62].

Current evidence indicates that maintaining relatively low ROS levels significantly improves resistance of postharvest litchi fruit to *P. litchii* [14,63]. Comparative studies reveal that the resistant cultivar ‘Heiye’ exhibits higher POD, SOD, and CAT activities than the susceptible cultivar ‘Guiwei’ [63]. Studies on various resistance-inducing treatments also reveal distinct regulatory patterns in the antioxidant system: *Exiguobacterium acetylicum* Strain SI17 enhances SOD activity while suppressing POD and CAT activities [5]; chitosan treatment upregulates SOD, CAT, and APX activities [17]; and 6-Pentyl-2H-pyran-2-one significantly enhances SOD and CAT activities along with DPPH scavenging activity while suppressing POD activity [18]. Moreover, postharvest applications of 6-benzylaminopurine [14], Tyr-Asp [64], *Bacillus amyloliquefaciens* LY-1 [65], and combined treatments with *Debaryomyces hansenii* and *Bacillus atrophaeus* [66], have been reported to improve antioxidant capacity, reducing decay and browning in litchi fruit.

The results of this study indicated that the resistance to litchi downy blight induced by ethephon treatment was closely associated with the regulation of ROS metabolism in postharvest litchi fruit. Although the O_2_^•−^ production rate in ethephon-treated fruit was significantly higher than that in the control at 48 and 96 h after inoculation, SOD activity was only significantly elevated at 24 h and significantly decreased by 96 h (Figure 2 and Figure 3). This suggested that the promotive effect of ethephon on H_2_O_2_ generation occurred mainly during the early storage period. Meanwhile, ethephon treatment also significantly enhanced the activities of CAT and APX, as well as DPPH scavenging capacity, thereby effectively reducing H_2_O_2_ accumulation and MDA content (Figure 3). These findings further supported that maintaining lower H_2_O_2_ levels contributed to enhanced resistance of litchi fruit to *P. litchii*. A similar mechanism was reported in ‘Kyoho’ grape, where ethephon application enhanced resistance to *Botrytis cinerea* by increasing *VvAPX* expression and elevating SOD and CAT activities [30]. In other studies, however, the antioxidant regulation mediated by ethephon (ethylene) in postharvest fruits has been extensively reported in the context of ripening [67], senescence [22], and chilling injury [68]. Furthermore, in wheat leaves, ethephon improves resistance against *Stagonospora nodorum* by enhancing POD and CAT activities while reducing H_2_O_2_ levels [27]. In contrast, activation of the ethylene signaling pathway in poplar and citrus leaves enhances disease resistance by promoting ROS production [69,70]. Collectively, these studies suggested that ethylene may induce disease resistance through species- or tissue-specific modulation of ROS dynamics.

In summary, ethephon application mitigated postharvest litchi downy blight by synergistically enhancing SOD, CAT, and APX activities and boosting DPPH scavenging capacity, thereby strengthening the antioxidant defense system of litchi fruit.

### 4.3. Ethephon Treatment Inhibited the Development of Litchi Downy Blight by Improving the Defense Capacity of Postharvest Litchi Fruit

CHI and GLU are key pathogenesis-related proteins that contribute to plant disease resistance by synergistically degrading pathogens cell walls [71]. PAL, as the rate-limiting enzyme of the phenylpropanoid pathway, plays a central role in the biosynthesis of defense compounds such as phenolics, lignin, and phytoalexins [72]. Elevated activities of CHI, GLU, and PAL have been demonstrated to enhance disease resistance in various postharvest fruits, including pitaya [73], peach [74], mango [75], and tomato [33].

In the resistance against postharvest litchi downy blight, CHI, GLU, and PAL also play equally critical roles. Upon infection by *P. litchii*, increased PAL activity promotes the accumulation of total phenolics, flavonoids, and anthocyanins, thereby strengthening non-enzymatic antioxidant capacity [4,11]. Additionally, lignin deposition has been shown to enhance resistance against *P. litchii* [63]. Notably, resistance induced by apple polyphenols [11], *Debaryomyces hansenii* and *Bacillus atrophaeus* [66] has been associated with increased GLU activity. Similarly, treatments with chitosan [17] and *Bacillus amyloliquefaciens* LY-1 [76] have been linked to upregulation of CHI, GLU, and PAL activity to reduce disease and decay in litchi fruit.

The results of this study demonstrated that ethephon treatment significantly enhanced the activities of CHI, GLU, and PAL, along with upregulating the expression of their key encoding genes in postharvest litchi fruit (Figure 4). These findings are consistent with observations across multiple plant species. Previous studies have shown that ethephon application activates the expression of *CHI* and *GLU* genes in tomato and grape fruits, thereby enhancing resistance to *Botrytis cinerea* [28,30]. Concurrently, in grape berries, ethephon-induced resistance is also closely associated with the activation of the phenylpropanoid pathway and the accumulation of flavonoids [30]. Similarly, during powdery mildew infection, genes including *CHIT4c*, *GLU*, and *PAL* in grapevines are upregulated by ethephon [77]. Furthermore, overexpression of key ethylene signaling transcription factors *SlERF1/2* in tomato markedly increases the activities of CHI, GLU, and PAL, consequently strengthening resistance against both *Rhizopus nigricans* and *Botrytis cinerea* [33,78]. The regulatory role of ethephon in the expression of these defense-related genes has also been validated in diverse plants such as melon [24], mandarins [29], and peach [79], indicating a conserved mechanism in ethylene-mediated disease resistance across species.

In all, ethephon likely strengthens resistance to litchi downy blight by upregulating the activities of key defense-related enzymes, namely CHI, GLU, and PAL.

## 5. Conclusions

In conclusion, this study provides the first demonstration that treatment with 400 mg·L^−1^ ethephon effectively controls postharvest litchi downy blight and elucidates its underlying mechanism in inducing disease resistance. The treatment significantly decreased both disease index and pericarp browning index while promoting substantial ethylene release. The released ethylene further enhanced DPPH scavenging capacity and increased the activities of SOD, CAT, APX, CHI, GLU, and PAL, as well as upregulating the expression of *LcCAT*, *LcAPX*, *LcCHI*, *LcGLU*, and *LcPAL*. These coordinated responses collectively improved the ROS scavenging capacity and strengthened disease resistance in litchi fruit. Our findings confirmed that ethephon treatment holds considerable potential for alleviating quality deterioration in postharvest litchi fruit caused by litchi downy blight.

## Figures and Tables

**Figure 1 foods-14-03493-f001:**
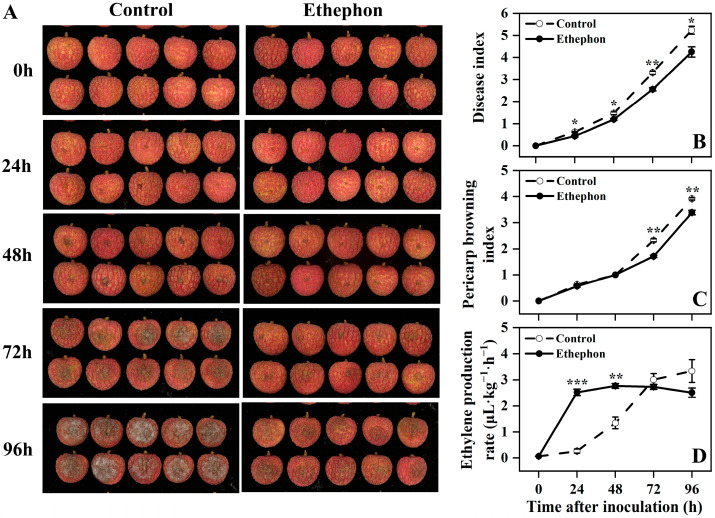
Effect of ethephon treatment on postharvest litchi appearance (**A**), disease index (**B**), pericarp browning index (**C**), and the rate of ethylene production (**D**) after inoculation with *P. litchi*. Asterisks in the graphs indicate significant differences between the control (water) and ethephon-treated group data at the same time point (* *p* < 0.05, ** *p* < 0.01, and *** *p* < 0.001).

**Figure 2 foods-14-03493-f002:**
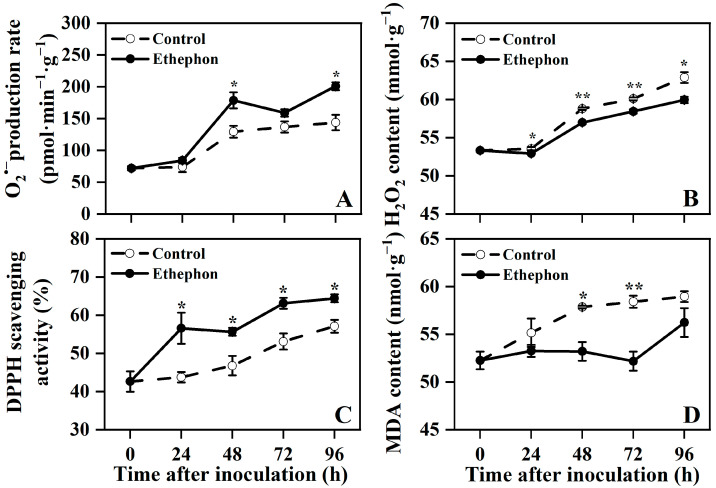
Effect of ethephon treatment on the superoxide anion (O_2_^•−^) production rate (**A**), hydrogen peroxide (H_2_O_2_) content (**B**), 2,2-Diphenyl-1-picrylhydrazyl (DPPH) scavenging activity (**C**), and malondialdehyde (MDA) content (**D**) in postharvest litchi fruits after inoculation with *P. litchi*. Asterisks in the graphs indicate significant differences between the control (water) and ethephon-treated group data at the same time point (* *p* < 0.05, ** *p* < 0.01).

**Figure 3 foods-14-03493-f003:**
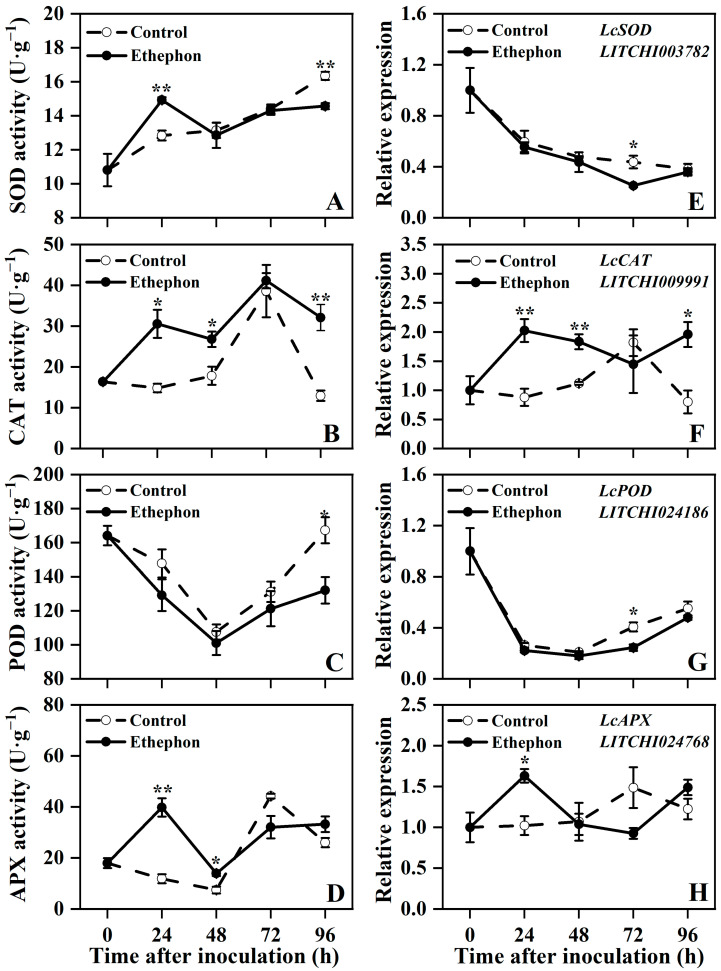
Effect of ethephon treatment on superoxide dismutase (SOD) activity (**A**), catalase (CAT) activity (**B**), peroxidase (POD) activity (**C**), ascorbate peroxidase (APX) activity (**D**), and their corresponding gene expression of *LcSOD* (**E**), *LcCAT* (**F**), *LcPOD* (**G**), *LcAPX* (**H**) in postharvest litchi fruits after inoculation with *P. litchi*. Asterisks in the graphs indicate significant differences between the control (water) and ethephon-treated group data at the same time point (* *p* < 0.05, ** *p* < 0.01).

**Figure 4 foods-14-03493-f004:**
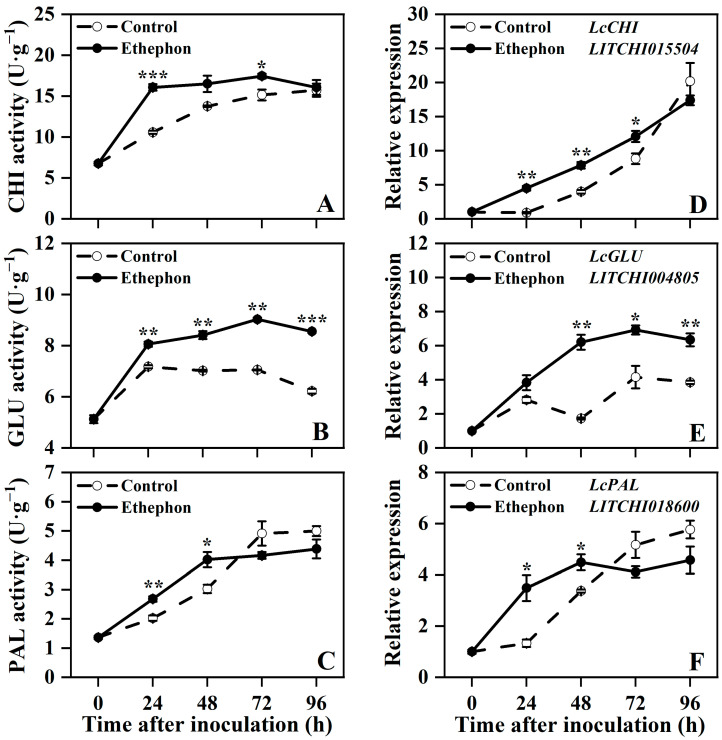
Effect of ethephon treatment on chitinase (CHI) activity (**A**), β-1,3-glucanase (GLU) activity (**B**), phenylalanine ammonia-lyase (PAL) activity (**C**), and their corresponding gene expression of *LcCHI* (**D**), *LcGLU* (**E**), *LcPAL* (**F**) in postharvest litchi fruits after inoculation with *P. litchi*. Asterisks in the graphs indicate significant differences between the control (water) and ethephon-treated group data at the same time point (* *p* < 0.05, ** *p* < 0.01, and *** *p* < 0.001).

**Figure 5 foods-14-03493-f005:**
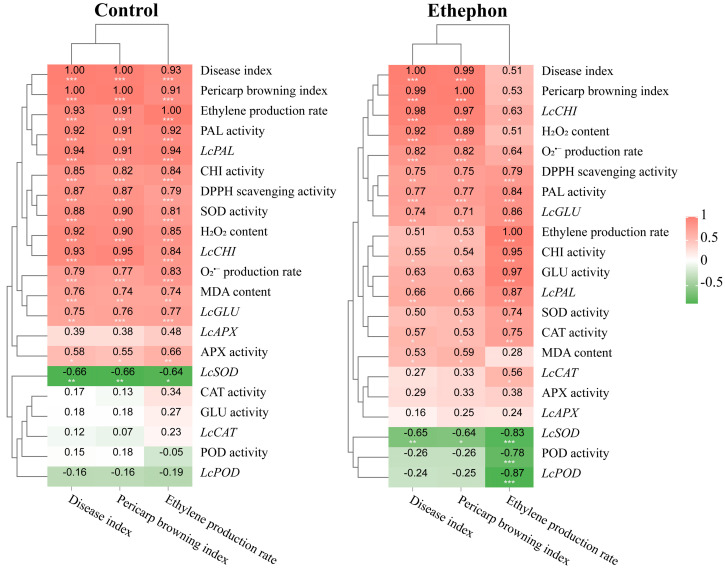
Correlation and hierarchical cluster analysis of physiological and biochemical indices in control (**A**) and ethephon-treated (**B**) fruits. Asterisks in the graphs indicate significant correlation (* *p* < 0.05, ** *p* < 0.01, and *** *p* < 0.001).

## Data Availability

The original contributions presented in this study are included in the article/Appendix A. Further inquiries can be directed to the corresponding author.

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
