# Peer review of "Ethephon Treatment Enhanced Postharvest Litchi Fruit Resistance to Peronophythora litchii by Strengthening Antioxidant Capacity and Defense Systems"

_foods, 2025, doi:10.3390/foods14203493_

Round 1
Reviewer 1 Report
Comments and Suggestions for Authors
The manuscript entitled “Ethephon treatment enhanced postharvest lichi fruit resistance to Peronophythora litchii by strengthening the antioxidant capacity and defense systems” is interesting. However, it is necessary to make some minor changes.
L108. Check the exponents that should be in superscript.
L111. Sterilized or disinfected?
L147. When do you use 0.1 and when do you use 0.01 in the change of absorbance to determine one enzymatic unit?
Please review the document to change "antioxidant activity" to "antioxidant capacity".
Author Response
Comments 1: L108. Check the exponents that should be in superscript.
Response 1: Thank you for pointing this out. We agree with this comment and have revised "1×104" to "1×104". The change can be found on page 3, subsection 2.1, line 111.
Comments 2: L111. Sterilized or disinfected?
Response 2: We thank you for this valuable correction. The term has been changed from "…sterilized using a 450 mg·L−1 prochloraz solution…" to "…disinfected using a 450 mg·L−1 prochloraz solution…" to accurately reflect that the litchi fruit were treated with 450 mg·L−1 prochloraz solution, which reduces microbial load but does not guarantee sterility. The correction can be found on page 3, subsection 2.2, line 113.
Comments 3: L147. When do you use 0.1 and when do you use 0.01 in the change of absorbance to determine one enzymatic unit?
Response 3: Thank you for pointing out this issue. We have corrected the definition based on the enzyme activity calculation formula: "For the definition of enzyme activity, one unit (U) of CAT, POD, or APX activity was defined as a change in absorbance of 0.1 per minute in the enzymatic reaction system." The correction can be found on page 5, paragraph 2, lines 199-200.
Comments 4: Please review the document to change "antioxidant activity" to "antioxidant capacity"
Response 4: We appreciate your valuable correction. We have carefully reviewed the manuscript and have revised "…antioxidant activity and pathogen defense capacity." to "…antioxidant and pathogen defense capacities.". The correction can be found on page 1, abstract, line 38.

Reviewer 2 Report
Comments and Suggestions for Authors
Comments to the author
The article “Ethephon Treatment Enhanced Postharvest Litchi Fruit Resistance to Peronophythora Litchii by Strengthening the Antioxidant Capacity and Defense Systems” presents a comprehensive study about the impact of ethephon treatment in the postharvest quality in litchi fruit after contamination by Peronophythora litchi. Since litchi downy blight is a serious disease that leads to substantial economic losses and food waste, this article proves to be an interesting topic. Ethephon treatment enhances litchi fruit resistance to Peronophythora litchii by regulating ethylene production, which in turn affects gene expression. This treatment helps maintain high levels of ethylene and increase activities of key antioxidant and defense-related enzymes, thereby improving the fruit's ability to resist infection. Overall, the manuscript is solid, with only a few aspects that I would like to comment on:
- In the tittle it is written “Peronophythora Litchii”. The upper-case letter in in the species name should be amended to a lower-case letter.
- In Material and Methods, the “Fruit Material and litchi” section would benefit from a more detailed description.
- Line 108 – “…containing 1×104 sporangia…” should read “…containing 1×104 sporangia…”.
- Line 132 – Why eight fruits only?
- Material and Methods subsections 2.5. “Determination of Reactive Oxygen Species (ROS) Levels, 2,2-Diphenyl-1-picrylhydrazyl (DPPH) scavenging activity and malondialdehyde (MDA) Contents” and 2.6. “Determination of Antioxidant Enzymes Activities” should have a broader description of the methods used and not only a reference for other studies.
- In figures 1 through 4, graphs must be further apart since scale units get merged. Also, for better clarity and interpretation, different line types (example: dash, dotted, etc.) should be used instead of different shaped markers on the line.
- Conclusion section should be better discussed to make clear of the novelty of the study as well as the impact of this research in the field.
My major concern about this study lies in the safety of the application of ethephon treatment for human consumption:
- Did the authors consider the potential toxicity or safety implications of ethephon residues for human consumption of treated litchi fruit? I think information should be included in the manuscript due to being an important aspect when it comes to the commercialization of the product, directly relating to human health.
- Did you analyze the ethephon residues on litchi at harvest, particularly in the edible portion (peeled aril), to determine the final concentration present in the fruit? If not, could residues potentially persist in the aril despite the peel being removed?

Author Response
Comments 1: In the tittle it is written “Peronophythora Litchii”. The upper-case letter in in the species name should be amended to a lower-case letter.
Response 1: Thank you for the correction. We have revised "Ethephon Treatment Enhanced Postharvest Litchi Fruit Resistance to Peronophythora Litchii by Strengthening the Antioxidant Capacity and Defense Systems" to "Ethephon Treatment Enhanced Postharvest Litchi Fruit Resistance to Peronophythora litchii by Strengthening the Antioxidant Capacity and Defense Systems" accordingly. The correction can be found on page 1, title, line 3.
Comments 2: In Material and Methods, the “Fruit Material and P. litchii” section would benefit from a more detailed description.
Response 2: Thank you for this valuable suggestion. We have supplemented the "Fruit Material and P. litchii" section with relevant details as followed:
“2.1. Fruit Material and P. litchii
Mature litchi fruits (cultivar ‘Huaizhi’) were collected in the morning from the Litchi Exposition Park in Guangzhou, Guangdong, China. After transport to the laboratory within three hours of harvest, healthy fruits with uniform size were selected and washed for the experiment.
The P. litchii isolate was provided by the College of Plant Protection, South China Agricultural University, and cultured on carrot agar media (200 g·L−1 carrot juice, 20 g·L−1 agar) at 25 °C for five days [39]. Following incubation, sporangia were eluted with sterile water and filtered to remove detached mycelia. A suspension adjusted to a concentration of 1×104 sporangia per mL was than prepared for inoculation.”
The correction can be found on page 3, subsection 2.1, lines 105, 109-111. Should further elaboration be needed, we would be pleased to provide additional information.
Comments 3: Line 108 – “…containing 1×104 sporangia…” should read “…containing 1×104 sporangia…”.
Response 3: Thank you for pointing this out. We agree with this comment and have revised "1×104" to "1×104". The change can be found on page 3, subsection 2.1, line 111.
Comments 4: Line 132 – Why eight fruits only?
Response 4: Thank you for the question. Since 'Huaizhi' litchi fruits are relatively small and each individual fruit releases only a limited amount of ethylene, using eight fruits for measurement ensures a more representative ethylene collection. Furthermore, this quantity covers the bottom of the sealed container, which helps maintain a stable environment for the measurement.
Comments 5: Material and Methods subsections 2.5. “Determination of Reactive Oxygen Species (ROS) Levels, 2,2-Diphenyl-1-picrylhydrazyl (DPPH) scavenging activity and malondialdehyde (MDA) Contents” and 2.6. “Determination of Antioxidant Enzymes Activities” should have a broader description of the methods used and not only a reference for other studies.
Response 5: Thank you for this suggestion. We have revised subsections 2.5 and 2.6 of the Materials and Methods to provide a more detailed description of the methods. The correction can be found on pages 4-5, lines 141-200.
Comments 6: In figures 1 through 4, graphs must be further apart since scale units get merged. Also, for better clarity and interpretation, diGerent line types (example: dash, dotted, etc.) should be used instead of diGerent shaped markers on the line.
Response 6: We appreciate your valuable comments. The figures (1-4) have been revised accordingly. Should you have any further suggestions, we would be happy to make additional changes. The correction can be found on page 6, line 249 (Figure 1); page 7, line 269 (Figure 2); page 8, line 293 (Figure 3); page 9, line 314 (Figure 4).
Comments 7: Conclusion section should be better discussed to make clear of the novelty of the study as well as the impact of this research in the field.
Response 7: We appreciate your insightful comment and have accordingly revised the Discussion and Conclusion sections to address this point. The specific modifications are as follows:
"Ethephon, as a low-toxicity, economical, and commercially available ethylene-releasing agent, plays a significant role in regulating plant disease resistance [23,24]. However, its application in postharvest litchi fruit preservation has not been previously reported. Our study established that treating postharvest litchi fruit with 400 mg·L-1 ethephon effectively suppressed downy blight development and significantly reduced both disease index and pericarp browning index (Figure 1).". The change can be found on page 10, subsection 4.1, lines 352 and 357.
" In conclusion, this study provided the first demonstration that treatment with 400 mg·L-1 ethephon effectively controlled postharvest litchi downy blight and elucidates its underlying mechanism in inducing disease resistance. …… Our findings confirmed that ethephon treatment holds considerable potential for alleviating quality deterioration in postharvest litchi fruit caused by litchi downy blight.". The change can be found on page 13, subsection 5, lines 465-467, 473-475.
Comments 8: Did the authors consider the potential toxicity or safety implications of ethephon residues for human consumption of treated litchi fruit? I think information should be included in the manuscript due to being an important aspect when it comes to the commercialization of the product, directly relating to human health.
Response 8: Thank you for this valuable comment. Potential safety concerns regarding the application of ethephon were carefully considered in our experimental design. First, in our preliminary concentration screening experiments, no significant difference was observed in the control efficacy of postharvest litchi downy blight between 400 mg·L-1 and 800 mg·L-1 ethephon. Therefore, the lower concentration of 400 mg·L⁻¹ was selected for subsequent investigations. Second, existing studies have shown that in postharvest mango and tomato fruits treated with 1000 mg·L-1 ethephon (by soaking for 5 minutes), the residue of ethephon can decrease to below 1 mg·kg-1 within 3 days [56,57], which is lower than the maximum residue limit (2 mg·kg-1) specified in the Chinese National Food Safety Standard (GB 2763-2021) for fruits such as mango, tomato, and litchi. Therefore, it can be reasonably concluded that the con-centration of ethephon used in this study poses a relatively low risk to food safety. We have added relevant explanations to the discussion section, and the specific additions are as follows:
"This concentration was determined through preliminary systematic screening experiments (conducted at 0, 200, 400, and 800 mg·L-1) with due consideration for food safety requirements.". The added content can be found on page 10, subsection 4.1, lines 357-359.
"Furthermore, existing studies have shown that in postharvest mango and tomato fruits treated with 1000 mg·L-1 ethephon (by soaking for 5 minutes), the residue of ethephon can decrease to below 1 mg·kg-1 within 3 days [56,57], which is lower than the maximum residue limit (2 mg·kg-1) specified in the Chinese National Food Safety Standard (GB 2763-2021) for fruits such as mango, tomato, and litchi. Therefore, it can be reasonably concluded that the concentration of ethephon used in this study poses a relatively low risk to food safety.". The change can be found on pages 10-11, subsection 4.1, lines 362 and 368.
Comments 9: Did you analyze the ethephon residues on litchi at harvest, particularly in the edible portion (peeled aril), to determine the final concentration present in the fruit? If not, could residues potentially persist in the aril despite the peel being removed?
Response 9: We appreciate your interest in our work. We would like to clarify that the current study did not directly measure ethephon residues in the pulp of postharvest litchi fruit. Although we previously noted that in mango and tomato fruits treated with 1000 mg·L-1 ethephon, the residue levels were below the limit specified in the Chinese National Food Safety Standard, and that the endocarp of litchi fruit is covered by a dense, hydrophobic waxy layer that may partially limit chemical penetration, we fully acknowledge the importance of food safety concerns raised. Following your suggestion, we plan to incorporate ethephon residue analysis and a systematic safety assessment into our future work to further evaluate the edible safety of this treatment.

Reviewer 3 Report
Comments and Suggestions for Authors
Thank you very much for the opportunity to review the manuscript. The work is timely, methodologically sound, and highly relevant. Only minor improvements are needed.
The study tested only a single ethephon concentration (400 mg/L), providing no dose–response data. This one-dose approach makes it unclear if lower or higher levels might be more effective or if 400 mg/L is optimal. It also raises questions about the generality of the findings, as no other concentrations or treatment durations were evaluated for comparison.
All fruits were inoculated with P. litchii after treatment. The absence of an uninfected control (fruits treated with ethephon but not inoculated) is a notable omission. Without a no-pathogen control, it’s impossible to discern whether ethephon alone (in the absence of disease) has any effect on pericarp browning, fruit senescence, or baseline enzyme levels. This is important because ethylene treatments can potentially accelerate ripening or browning in fruits; the manuscript does not address whether ethephon might adversely affect litchi quality on its own.
Some of the results are internally inconsistent or counter-intuitive and are not adequately explained. For example, ethephon-treated fruits showed higher superoxide anion production (O₂·⁻) at certain times (48 and 96 h) compared to controls, yet simultaneously had lower hydrogen peroxide (H₂O₂) levels. On the surface this seems contradictory – one would expect elevated superoxide generation to produce more H₂O₂ (via dismutation). The manuscript attributes reduced H₂O₂ to improved scavenging, but does not clearly reconcile how O₂·⁻ was higher in treated fruits (it merely states this observation without mechanistic explanation). This raises questions about the interpretation: why is superoxide generation elevated under a treatment meant to reduce oxidative damage? The discussion touches on “species- or tissue-specific modulation of ROS” (lines 680–688) but fails to give a satisfying explanation. A reviewer would likely flag the need to clarify this apparent paradox and possibly measure or discuss the role of SOD activity in converting O₂·⁻ to H₂O₂ (since SOD activity was modulated by ethephon).
The use of Pearson correlation and clustering (Figure 5) to identify “critical” enzymes is not entirely convincing. The authors observe that in ethephon-treated fruits, ethylene production correlates more strongly with CAT, GLU, and CHI activities (and LcCAT expression) than in controls, and conclude that “CAT, GLU, and CHI played more critical roles in ethephon-induced disease resistance”. This is somewhat speculative: correlation does not equal causation. Just because those enzymes correlate with ethylene or disease outcomes under treatment does not prove they are the key drivers of resistance. The manuscript lacks any functional evidence (such as enzyme inhibitors or knockout studies) to substantiate that these particular enzymes are essential. A careful reviewer would caution that the authors are over-interpreting correlative data. At minimum, the text should acknowledge this is an association. As is, the claims about “more critical roles” are too strong given the data basis.
The novelty and distinctiveness of the results are not clearly highlighted in relation to previous studies.
The manuscript contains several minor mistakes that need correction to meet publication standards. In the Materials and Methods, section numbering is confused. “2.5” is used twice (once for ROS/DPPH/MDA determinations and again for RNA extraction/qPCR), which appears to be a typographical error. There are a few typos and English errors (e.g., “PAL activaty” instead of activity in line 714, and referring to “diverse plant such as melon” – which should be plural “plants”). Figure references in the text are sometimes formatted incorrectly. These should be carefully edited to meet the journal’s quality standards.
Author Response
Comments 1: The study tested only a single ethephon concentration (400 mg/L), providing no dose–response data. This one-dose approach makes it unclear if lower or higher levels might be more effective or if 400 mg/L is optimal. It also raises questions about the generality of the findings, as no other concentrations or treatment durations were evaluated for comparison.
Response 1: Thank you for this valuable comment. Prior to this study, we systematically evaluated the inhibitory effects of different concentrations of ethephon (0, 200, 400, and 800 mg/L) on postharvest litchi downy blight. The screening results demonstrated that the 400 mg/L ethephon treatment significantly suppressed disease development, with no statistically significant difference in efficacy compared to the higher concentrations (800 mg/L). Furthermore, we confirmed that a 2-minute immersion in 400 mg/L ethephon significantly reduced the infection of postharvest 'Feizixiao' and 'Huaizhi' litchi fruit by P. litchii. Based on both food safety considerations and practical applicability, we ultimately selected the lower concentration of 400 mg/L combined with a 2-minute immersion time in order to focus on elucidating the mechanism of ethephon-induced resistance. We have added relevant explanations to the discussion section, and the specific additions are as follows:
"Our study established that treating postharvest litchi fruit with 400 mg·L-1 ethephon effectively suppressed downy blight development and significantly reduced both disease index and pericarp browning index (Figure 1). This concentration was determined through preliminary systematic screening experiments (conducted at 0, 200, 400, and 800 mg·L-1) with due consideration for food safety requirements.". The added content can be found on page 10, subsection 4.1, lines 355-359.
Comments 2: All fruits were inoculated with P. litchii after treatment. The absence of an uninfected control (fruits treated with ethephon but not inoculated) is a notable omission. Without a no-pathogen control, it’s impossible to discern whether ethephon alone (in the absence of disease) has any effect on pericarp browning, fruit senescence, or baseline enzyme levels. This is important because ethylene treatments can potentially accelerate ripening or browning in fruits; the manuscript does not address whether ethephon might adversely affect litchi quality on its own.
Response 2: Thank you for pointing this out. Previous studies have indicated that the role of ethephon (ethylene) in promoting fruit ripening and senescence is primarily observed in climacteric fruits, such as bananas, where the ripening process is accompanied by a decline in disease resistance, leading to disease outbreaks and tissue browning and decay. Since litchi is a non-climacteric fruit, the potential direct effects of ethephon on non-inoculated fruit were not initially considered in our experimental design. We fully agree with the reviewer's perspective and will conduct a systematic analysis of this aspect in subsequent work to more comprehensively evaluate the role of ethephon in postharvest litchi preservation. We greatly appreciate the reviewer's guidance and welcome any further suggestions for our research.
Comments 3: Some of the results are internally inconsistent or counter-intuitive and are not adequately explained. For example, ethephon-treated fruits showed higher superoxide anion production (O₂·−) at certain times (48 and 96 h) compared to controls, yet simultaneously had lower hydrogen peroxide (H₂O₂) levels. On the surface this seems contradictory – one would expect elevated superoxide generation to produce more H₂O₂ (via dismutation). The manuscript attributes reduced H₂O₂ to improved scavenging, but does not clearly reconcile how O₂·− was higher in treated fruits (it merely states this observation without mechanistic explanation). This raises questions about the interpretation: why is superoxide generation elevated under a treatment meant to reduce oxidative damage? The discussion touches on “species- or tissue-specific modulation of ROS” (lines 680–688) but fails to give a satisfying explanation. A reviewer would likely flag the need to clarify this apparent paradox and possibly measure or discuss the role of SOD activity in converting O₂·− to H₂O₂ (since SOD activity was modulated by ethephon).
Response 3: Thank you for this valuable comment. We have, accordingly, provided a more detailed explanation in the Discussion section regarding the differential changes in O₂·⁻ production rate and H₂O₂ content in ethephon-treated fruits. The specific additions are as follows:
"The results of this study indicated that the resistance to litchi downy blight induced by ethephon treatment was closely associated with the regulation of ROS metabolism in postharvest litchi fruit. Although the O₂·− production rate in ethephon-treated fruit was significantly higher than that in the control at 48 and 96 hours after inoculation, SOD activity was only significantly elevated at 24 hours and significantly decreased by 96 hours (Figure 2 & 3). This suggested that the promotive effect of ethephon on H₂O₂ generation occurred mainly during the early storage period. Meanwhile, ethephon treatment also significantly enhanced the activities of CAT and APX, as well as DPPH scavenging capacity, thereby effectively reducing H₂O₂ accumulation and MDA content (Figure 3). These findings further supported that maintaining lower H₂O₂ levels contributed to enhanced resistance of litchi fruit to P. litchii.". The added content can be found on pages 11-12, subsection 4.2, lines 405-415.
Comments 4: The use of Pearson correlation and clustering (Figure 5) to identify “critical” enzymes is not entirely convincing. The authors observe that in ethephon-treated fruits, ethylene production correlates more strongly with CAT, GLU, and CHI activities (and LcCAT expression) than in controls, and conclude that “CAT, GLU, and CHI played more critical roles in ethephon-induced disease resistance”. This is somewhat speculative: correlation does not equal causation. Just because those enzymes correlate with ethylene or disease outcomes under treatment does not prove they are the key drivers of resistance. The manuscript lacks any functional evidence (such as enzyme inhibitors or knockout studies) to substantiate that these particular enzymes are essential. A careful reviewer would caution that the authors are over-interpreting correlative data. At minimum, the text should acknowledge this is an association. As is, the claims about “more critical roles” are too strong given the data basis.
Response 4: We are grateful that you raised this point and are in complete agreement with your view. Accordingly, we have adjusted the expression from "Overall, these results collectively suggested that CAT, GLU, and CHI played more critical roles in ethephon-induced disease resistance." to "Taken together, these findings indicated that changes in the activities of CAT, GLU, and CHI in response to ethephon treatment exhibited a strong association with ethylene production.". The correction can be found on page 9, subsection 3.5, lines 332-334. We would welcome it if you could kindly provide further suggestions should this wording require additional improvement.
Comments 5: The novelty and distinctiveness of the results are not clearly highlighted in relation to previous studies.
Response 5: We appreciate your insightful comment and have accordingly revised the Discussion and Conclusion sections to address this point. The specific modifications are as follows:
"Ethephon, as a low-toxicity, economical, and commercially available ethylene-releasing agent, plays a significant role in regulating plant disease resistance [23,24]. However, its application in postharvest litchi fruit preservation has not been previously reported. Our study established that treating postharvest litchi fruit with 400 mg·L-1 ethephon effectively suppressed downy blight development and significantly reduced both disease index and pericarp browning index (Figure 1).". The change can be found on page 10, subsection 4.1, lines 352 and 357.
" In conclusion, this study provided the first demonstration that treatment with 400 mg·L-1 ethephon effectively controlled postharvest litchi downy blight and elucidates its underlying mechanism in inducing disease resistance. …… Our findings confirmed that ethephon treatment holds considerable potential for alleviating quality deterioration in postharvest litchi fruit caused by litchi downy blight.". The change can be found on page 13, subsection 5, lines 465-467, 473-475.
Comments 6: The manuscript contains several minor mistakes that need correction to meet publication standards. In the Materials and Methods, section numbering is confused. “2.5” is used twice (once for ROS/DPPH/MDA determinations and again for RNA extraction/qPCR), which appears to be a typographical error. There are a few typos and English errors (e.g., “PAL activaty” instead of activity in line 714, and referring to “diverse plant such as melon” – which should be plural “plants”). Figure references in the text are sometimes formatted incorrectly. These should be carefully edited to meet the journal’s quality standards.
Response 6: Thank you for pointing these out. We have made the following revisions as suggested:
- We have corrected the numbering of the subsections "2.8. RNA Extraction and Gene Expression Analysis" and "2.9. Statistical Analyses". The change can be found on page 5, subsections 2.8 and 2.9, lines 211 and 224.
- We have revised "…PAL activaty…" to "…PAL activity…". The change can be found on page 12, subsection 4.3, line 438.
- We have revised "…diverse plant such as melon…" to "…diverse plants such as melon…". The change can be found on page 12, subsection 4.3, line 459.
- We have revised "Fig. 5B" to " Figure 5B ". The change can be found on page 9, subsection 3.5, line 332.
- We have revised "Figures 1 & 5" to "Figure 1 & 5". The change can be found on page 10, subsection 4.1, line 346.
- We have revised "Figures. 1" to "Figure 1". The change can be found on page 11, paragraph 2, line 380.

Round 2
Reviewer 2 Report
Comments and Suggestions for Authors
Dear, Authors,
Thank you for carefully addressing the comments from the major revision. I am pleased to confirm that your revised manuscript has satisfactorily resolved the concerns raised. I therefore recommend acceptance of the paper for publication.